# Mutation-Based Antibiotic Resistance Mechanism in Methicillin-Resistant *Staphylococcus aureus* Clinical Isolates

**DOI:** 10.3390/ph14050420

**Published:** 2021-05-01

**Authors:** Tanveer Ali, Abdul Basit, Asad Mustafa Karim, Jung-Hun Lee, Jeong-Ho Jeon, Shafiq ur Rehman, Sang-Hee Lee

**Affiliations:** 1Institute of Microbiology and Molecular Genetics, University of the Punjab, Lahore 54590, Pakistan; tanveer.alituri93@gmail.com (T.A.); basitbch@gmail.com (A.B.); 2Department of Bioscience and Biotechnology, The University of Suwon, Hwaseong, Gyeonggido 18323, Korea; asadmustafa8@gmail.com; 3National Leading Research Laboratory, Department of Biological Sciences, Myongji University, 116 Myongjiro, Yongin, Gyeonggido 17058, Korea; topmanlv@hanmail.net (J.-H.L.); najashin@hanmail.net (J.-H.J.)

**Keywords:** mutation, *mecA*, methicillin-resistant *Staphylococcus aureus*, allosteric site, PBP2a

## Abstract

β-Lactam antibiotics target penicillin-binding proteins and inhibit the synthesis of peptidoglycan, a crucial step in cell wall biosynthesis. *Staphylococcus aureus* acquires resistance against β-lactam antibiotics by producing a penicillin-binding protein 2a (PBP2a), encoded by the *mecA* gene. PBP2a participates in peptidoglycan biosynthesis and exhibits a poor affinity towards β-lactam antibiotics. The current study was performed to determine the diversity and the role of missense mutations of PBP2a in the antibiotic resistance mechanism. The methicillin-resistant *Staphylococcus aureus* (MRSA) isolates from clinical samples were identified using phenotypic and genotypic techniques. The highest frequency (60%, 18 out of 30) of MRSA was observed in wound specimens. Sequence variation analysis of the *mecA* gene showed four amino acid substitutions (i.e., E239K, E239R, G246E, and E447K). The E239R mutation was found to be novel. The protein-ligand docking results showed that the E239R mutation in the allosteric site of PBP2a induces conformational changes in the active site and, thus, hinders its interaction with cefoxitin. Therefore, the present report indicates that mutation in the allosteric site of PBP2a provides a more closed active site conformation than wide-type PBP2a and then causes the high-level resistance to cefoxitin.

## 1. Introduction

Infections by methicillin-resistant *Staphylococcus aureus* (MRSA) are considered a pivotal global health issue. The MRSA isolate first emerged in the United Kingdom in 1961 and quickly spread globally [1]. MRSA is able to spread both in hospitals and within the community [2,3]. *S. aureus* infects skin glands, intact skin, nasal cavity, and intestinal mucosa, which can lead to bacteremia, pneumonia, or wound and bone infections [3,4,5]. MRSA acquires resistance against β-lactam antibiotics via a penicillin-binding protein 2a (PBP2a) [6]. PBP2a (a monofunctional transpeptidase) has a low affinity for β-lactam antibiotics and thus cannot be inhibited by these antibiotics [6]. MRSA clinical isolates become resistant to a number of antibiotic classes (e.g., fluoroquinolones, macrolides, aminoglycosides, and clindamycin) and β-lactam antibiotics [3]. These additional resistance mechanisms are due to mutations and acquired resistance determinants, along with the formation of biofilm. Therefore, the MRSA isolates are multidrug-resistant (MDR) pathogens, which is a major cause of mortality and morbidity globally [3]. In MRSA, PBP2a shows catalytic activity in peptidoglycan biosynthesis (eventually, biosynthesis of the MRSA cell wall) in cooperation with the transglycosylase activity of other PBPs, such as PBP2 [1,7,8]. The peptidoglycan polymer consists of glycan strands comprised of a repeating disaccharide unit (N-acetylglucosamine (GlcNAc) and N-acetylmuramylpentapeptide (MurNAc pentapeptide): GlcNAc–MurNAc pentapeptide) [1]. The adjacent glycan strands are cross-linked by the transpeptidase activity of PBP2a using peptide stems present on each MurNAc saccharide. The cell wall encases the entire MRSA in a single molecule, and its integrity is essential to the survival of MRSA. The PBP2a is encoded by the *mecA* gene carried by a mobile genetic element, and the staphylococcal cassette chromosome *mec* (SCC*mec*) is present on the chromosomes of MRSA strains [9]. Currently, thirteen allotypes of SCC*mec* (namely type I–XIII) have been defined. Among the thirteen SCC*mec* allotypes, the only rare allotype (type XI) contains the *mecC* gene instead of the *mecA* gene [3]. Furthermore, PBP2x shows cefuroxime resistance in *Streptococcus pneumoniae* [10]. Generally, the *mecA* gene is considered an essential genotypic characteristic of MRSA and the target gene for rapid diagnosis of MRSA infections [11] while the *nuc* gene (*S. aureus*-specific chromosomal gene encoding thermonuclease) is used for the rapid detection of overall *S. aureus* (both MSSA (methicillin-sensitive *S. aureus*) and MRSA) from clinical samples [12].

Point mutations in the *mecA* gene likely affect the function of the mecA-encoded PBP2a, leading to a change in the methicillin resistance activity [10]. Different previous studies have reported mutations in *mecI* and *mecA*, but few data were able to establish the correlation between mutations in such genes and resistance to β-lactam antibiotics [13]. Ceftaroline, a recently introduced anti-MRSA β-lactam antibiotic, binds noncovalently to the allosteric site of PBP2a, which creates a conformational change to allow for opening of the active site. Then, a second ceftaroline binds to the now opened active site, thus inhibiting the transpeptidase function of PBP2a [1,14]. However, mutations in the allosteric site disrupts the allosteric opening and may play a vital role in creating resistance against ceftaroline. Previously, mutations in the allosteric site (N146K, E150K, and E239K) of PBP2a have been shown to have low-level resistance against ceftaroline [14,15]. Another study reported the N146K-N204K-G246E triple mutant having high rates of resistance to anti-MRSA β-lactam antibiotics such as ceftaroline and ceftobiprole [16]. A high-level of resistance to ceftaroline was observed in the MRSA isolates carrying the Y446N-E447K double mutation in the active site region of PBP2a [17].

There is very limited data available on the *mecA* genetic polymorphisms, particularly, in the allosteric site of the PBP2a in clinical MRSA isolates showing high-level resistance to cefoxitin. In the current study, we investigated the genetic polymorphism of the *mecA* gene in clinical MRSA isolates from different specimens of district Lahore, Pakistan. We further determined the probable role of the observed novel mutation in the allosteric site of PBP2a in antibiotic resistance by protein modeling and protein-ligand docking.

## 2. Results

### 2.1. Characteristics of S. aureus Isolates

In the current study, 33 *S. aureus* were phenotypically isolated from clinical samples (blood, nasal swab, wound, etc.) from Lahore (isolation sites: Main Boulevard, Zarar Shaheed Road, Lahore Road, and Link Road), Pakistan. These 33 isolates were *nuc*-positive and were further confirmed as *S. aureus* through 16 s ribotyping.

### 2.2. Characteristics of MRSA Isolates

Of the 33 *S. aureus*, 30 isolates were found to be resistant to cefoxitin (Table 1) and to be *mecA*-positive. Therefore, these 30 isolates are MRSAs because polymerase chain reaction (PCR) amplification of the *mecA* gene and cefoxitin resistance determined by disk diffusion tests are considered a gold standard method for the rapid detection of MRSA infections [18,19]. Of the 30 MRSA, three isolates (MR-13, -14, and -33) demonstrated high-level cefoxitin resistance with inhibition zone diameters ranging from 3 mm to 4 mm (Table 1) and high minimum inhibitory concentrations (MICs) (≥64 mg/L) (Table 2). The remaining seventeen MRSA isolates revealed low-level cefoxitin resistance with inhibition zone diameters ranging from 13 mm to 20 mm (Table 1).

The prevalence of MRSA from the different clinical samples was different. The highest frequency (60%, 18 out of 30) of MRSA was observed in wound specimens. Furthermore, the frequencies of MRSA from blood specimens, nasal swabs, and ear swabs were 23% (*n* = 7), 10% (*n* = 3), and 7% (*n* = 2).

### 2.3. Genetic Polymorphism of mecA Gene

To find genetic polymorphism in the *mecA* gene, the *mecA* coding sequences of the MRSA isolates were subjected to alignment with a reference sequence from MRSA strain N315. The multiple DNA sequence alignment (Figure 1a,b) showed nucleotide substitutions at five different positions (i.e., 75, 715, 716, 737, and 1339). All of the mutations were responsible for the amino acid substitutions except for a silent mutation at position 75 (Figure 1).

The amino acid substitutions (E239K, E239R, and G246K) took place at the active site of PBP2a and the fourth amino acid (E447K) was replaced at the allosteric site. The mutations (E239K, G246K, and E447K) from the three MRSA isolates (MR-13, -14, and 33) showing cefoxitin resistance with high MICs (≥64 mg/L) (Table 2) were reported to be involved in mediating the resistance to ceftaroline [15,16,17]. However, the E239R mutation (with additional E447K) demonstrating high-level resistance to cefoxitin (Table 2) is reported for the first time in this study. Therefore, the structural features of E239R in mediating the high-level resistance were investigated.

### 2.4. Structural Perspective of PBP2a with E239R Mutation on Cefoxitin Resistance

In order to assess the role of missense mutations in the structure and function (cefoxitin resistance) of PBP2a, we performed protein-ligand docking. The amino acid substitutions at positions 239 and 246 lie in the allosteric site of the non-penicillin-binding domain (allosteric domain), while E447K mutation exits from the active site of the transpeptidase domain of PBP2a [14]. Y446 and E447 lie within the active site of PBP2a [21]. Therefore, the Y446N and E447K mutants mediate resistance to ceftaroline [14,17]. However, the structural role of the mutation at E239 located in the allosteric site of PBP2a has not been clearly explained. The protein-ligand docking results showed that cefoxitin interacts directly with E239 at the allosteric site of wide-type PBP2a (Figure 2a). Furthermore, E239 is shown to be directly involved in the interaction with peptidoglycan (Figure 2c). However, the mutant PBP2a with E239R mutation has not shown any binding interactions with cefoxitin, as shown in Figure 2b. Furthermore, a ligand such as cefoxitin is positioned at E239 in the wild-type PBP2a (Figure 2d), while it is positioned away from R239 in the mutant PBP2a (Figure 2e). The mutation of glutamate at position 239 into arginine results in changing negatively charged side chains into positively charged R-groups. These results suggest that E239R mutation decreases the binding affinity of cefoxitin for arginine at the corresponding position and thus confers resistance to cefoxitin. The docking results showed that the binding affinity (−4.1 kcal/mol, the free energy of binding) of cefoxitin for the E239R mutant is lower than the binding affinity (−5.4 kcal/mol) of cefoxitin for the wild-type PBP2a, which indicates that cefoxitin has a lower affinity for the mutant PBP2a.

## 3. Discussion

All 33 *S. aureus* clinical isolates demonstrated successful amplification of the *nuc* gene and were further confirmed as *S. aureus* through 16s ribotyping, which indicates high specificity for the rapid PCR detection of *S. aureus* through amplifying the *nuc* gene. The other reports were in accordance with our observations of the high specificity of the *nuc* gene in *S. aureus* for identification purposes [12,22]. In addition, all 30 *mecA*-positive MRSA isolates were found to be resistant to cefoxitin. These data suggest that the cefoxitin disk-diffusion (CFD) method is a sensitive phenotypic testing used for the identification of MRSA, as previously reported [18]. This CDF method is able to avoid misdiagnosis that could be the significant factor for the emergence of infections with MRSA [23] and could lead to the development of resistance against other important available antibiotics. A study from Pakistan reported that 22% of the clinical isolates were misdiagnosed as MRSA [24]. Many factors influence the accuracy of the MRSA diagnosis: symptomology, type of laboratory test used, and the effectiveness of the utilized tests [25]. Such a misdiagnosis emphasizes the use of specific, rapid, and sensitive techniques for epidemiological and therapeutic studies.

The highest frequency of MRSA was observed in wound specimens. Similar studies from Pakistan and India have reported the highest prevalence of MRSA from wound specimens [26,27]. The highest frequency of MRSA from wounds could be due to its presence on the skin as normal flora. The frequency of MRSA from blood specimens was 23% (n = 7), which is greater than the previously reported study showing that 12% MRSA are found in blood specimens, followed by at the wound source (26%) [28]. The frequency of MRSA from nasal swabs was 10% (n = 3), which is in good agreement with the previously reported data from Pakistan [29]. The frequency of MRSA from ear swabs was 7% (n = 2), which is lower than the frequencies (27–41%) that were reported previously [29,30]. However, the sample size was so small that we were not able to draw a clear epidemiological aspect of MRSA showing the resistance to cefoxitin in Pakistan.

In our study, four different missense mutations were detected within PBP2a in MRSA MR-13, MR-14, and MR-33 clinical isolates. Three of the four missense mutations have been described as a contributor of antibiotic resistance in other countries: E239K (Spain [15], Thailand [15], and the United States [17]), G246E (Africa [16], Greece [15], and Switzerland [31]), and E447K (the United States [17]). As far as we know, the E239R missense mutation has not been reported yet, but seems to be present in Pakistan. Our data showed that the E239R-E447K double mutant rendered MRSA MR-33 highly resistant to cefoxitin (MIC of 128 mg/L). The MRSA isolates harboring the Y446N-E447K double mutation at the active site region of PBP2a showed high-level resistance (MIC > 32 mg/L) to ceftaroline [17]. It was suggested that the Y446N mutation in the Y446N-E447K double mutant PBP2a is the main contributor of the high-level resistance [17], which agrees with the previous structural studies that implicated Y446N as the gatekeeper to the active site of the transpeptidase domain [14]. Furthermore, the E447K mutation may be a minor contributor stabilizing the Y446N-E447K double mutant (or facilitating the transpeptidation reaction) [17]. According to these results, the most likely cause of the high-level resistance to cefoxitin may be the E239R mutation in the E239R-E447K double mutant PBP2a, although isogenic mutants were not tested to confirm this suggestion.

It has been shown that the PBP2a-mediated β-lactam resistance in MRSA is imparted by the weaker binding of β-lactam antibiotics to the closed PBP2a transpeptidase (TP) pocket (the closed active site) in the TP domain [14,15]. Therefore, PBP2a allows MRSA to maintain TP activity in the presence of β-lactam antibiotics. The closed active site is why 17 MRSA clinical isolates showed low-level cefoxitin resistance with inhibition zone diameters ranging from 13 mm to 20 mm. However, PBP2a relies on PBP2 for the transglycosylase (TG) activity because it does not have any TG activity [15]. The experimental evidence demonstrated that cooperative functions between PBP2a and PBP2 exist in MRSA peptidoglycan biosynthesis and that these cooperative functions are achieved through a direct protein–protein interaction between PBP2a and PBP2 [15]. The structural studies identified an allosteric site 60 Å away from the active site of PBP2a [14]. Ceftaroline (or peptidoglycan) binds to the allosteric site, which is consistent with the structural data of *S. pneumoniae* PBP2x [10]. When the allosteric site of PBP2a is bound by the ligand such as ceftaroline (or peptidoglycan), a conformational change allows for opening of the active site (allosteric opening), which permits ligand (ceftaroline) entry, facilitates ceftaroline binding at the active site, and eventually inhibits TP activity [14,17].

The missense mutations at the allosteric site of PBP2a impart three traits on the mutant PBP2a in MRSA clinical isolates showing resistance to ceftaroline [1,14]: (1) these mutations not only created new salt-bridges (not present in the wild-type PBP2a) around the mutated residues but also established a new salt-bridge network among many residues 35 Å away from the mutated positions (but the salt-bridge observed in the wild-type PBP2a is not detected in the mutant PBP2a); (2) missense mutations change the electrostatic potential in the allosteric site, showing a marked basic character extending beyond the immediate position of the mutated residues; and (3) these mutations decrease the second-order rate constant for the acylation in the active site, which indicates that the acylation event becomes more unfavorable. Due to these structural traits (alterations), the missense mutations at the allosteric site of PBP2a disrupt the allosteric opening and these mutations create a more closed active site conformation, which may play a vital role in creating high-level resistance to cefoxitin and ceftaroline. Furthermore, this antibiotic resistance mechanism is consistent with the protein-ligand docking data of the E239R mutant in MRSA isolates, which shows high-level resistance to cefoxitin.

In conclusion, the CDF method along with PCR amplification of the *mecA* gene may be a sensitive technique for the detection of MRSA infections. The E239R mutation at the allosteric site of PBP2a results in the unfavorable positioning of the amino acid R-group for binding to cefoxitin at the corresponding position. Therefore, the E239R mutation provides the favorable conformation of PBP2a for the interaction with peptidoglycan even in the presence of cefoxitin with a lower affinity for the mutant PBP2a. Overall, a missense mutation at the allosteric site of PBP2a plays a vital role in providing antibiotic resistance to MRSA. These results can be useful both in the understanding of the development of antibiotic resistance and the design of new anti-MRSA antibiotics.

## 4. Materials and Methods

### 4.1. Bacterial Isolates

A total of 33 *S. aureus* isolates were isolated from the clinical samples of the Citi Lab and Research Centre (Medical Laboratory, Lahore, Pakistan) in 2019. The confirmed MRSA isolates were stored at −80 °C in 20% glycerol. The MRSA isolates were cultivated on nutrient agar (Oxoid, Hampshire, UK) and mannitol salt agar (Oxoid, Hampshire, UK).

### 4.2. Identification of S. aureus from Clinical Samples

*S. aureus* was isolated phenotypically from the clinical samples using a previously reported method [32,33]. Furthermore, the molecular detection of *S. aureus* was performed by PCR amplification of the *nuc* gene with Fnuc and Rnuc primers (Table 3) [34]. The expected PCR product of 270 bp was verified by agarose gel electrophoresis (Appendix A). *S. aureus* harboring the *nuc* gene was subjected to PCR amplification of the 16Sr DNA fragment using 16sF and 16sR primers (Table 3). The amplified PCR products were sequenced using a previously reported method [35]. The DNA sequence analysis was carried out through MEGA-X (www.megasoftware.net; accessed on 29 April 2021) and BLAST (https://blast.ncbi.nlm.nih.gov/Blast.cgi; accessed on 29 April 2021).

### 4.3. Antimicrobial Susceptibility Testing

The antimicrobial susceptibility was tested by the Clinical Laboratory Standards Institute (CLSI) disk-diffusion method [20] on Müeller–Hinton agar with cefoxitin 30 µg disks (Oxoid). The plates were incubated for 24 h at 35 °C before measuring the inhibition zone diameters. *S. aureus* ATCC 25923 was included as a control strain. The inhibition zone diameters were interpreted using the CLSI criteria [20]. In the disk-diffusion method performed using a 30 µg cefoxitin disc, an inhibition zone diameter is considered methicillin resistant when ≤21 mm and is reported as methicillin sensitive when ≥22 mm. All MRSA isolates that demonstrated resistance to cefoxitin (Table 1) were further characterized as indicated below. Furthermore, MIC was determined by the broth microdilution method [37] for three MRSA isolates containing amino acid substitutions in PBP2a (Table 2).

### 4.4. Molecular Detection of mecA Gene

All of the MRSA isolates were confirmed by *mecA* gene detection through PCR amplification using mecA_DF and mecA_DR primers (Table 3). These forward and reverse primers were designed to amplify a highly conserved fragment of the *mecA* gene. The DNA was extracted using the QIAamp DNA Mini Kit (QIAGEN, Valencia, CA, USA), in accordance with the manufacturer’s protocol. The PCR amplifications were carried out on a DNA thermal cycler (mod. 2400, Perkin–Elmer Cetus, Norwalk, CT, USA) as previously described [38]. The DNA isolated from the MRSA isolates was subjected to PCR using the following conditions: initial denaturation for 4 min at 94 °C, followed by 30 cycles of denaturation at 94 °C for 30 s, annealing at 55 °C for 30 s, and extension for 1 min at 72 °C. Final extension was performed for 15 min at 72 °C. The expected PCR product of 533 bp was verified by agarose gel electrophoresis (Appendix A).

### 4.5. Genetic Polymorphism of mecA Gene

In order to determine the genetic polymorphism in the *mecA* gene, the entire *mecA* region including the *mecA* coding sequence was amplified from the MRSA isolates using MF and MR primers (Table 3). The primers were designed to amplify the complete gene sequence of the *mecA* gene from the complete genome sequences of MRSA (GenBank accession numbers CP039164 and NC_002745). The PCR amplification was performed with an initial denaturation at 94 °C for 5 min and 30 cycles of denaturation at 94 °C for 30 s, annealing at 56 °C for 30 s, and extension at 72 °C for 2 min. The final extension was performed at 72 °C for 20 min. The amplicon (2368 bp, Appendix A) was sequenced using the previously reported primers [36] and the newly designed primers (Table 3). The sequences were analyzed through MEGA-X by using the *mecA* gene (GenBank accession no. NC_002745 of MRSA strain N315) as a reference sequence [16].

### 4.6. Protein Modeling and Protein-Ligand Docking

In order to determine the structural change in mutant PBP2a caused by missense mutations found in the *mecA* gene of the MR-33 isolate, three-dimensional modeling of the mutant PBP2a carrying missense mutations (E239R and E447K) was performed by an automated homology modeling approach using the wild-type PBP2a (Protein Data Bank identification number ((PDB ID), 5M18) as a template in the SWISS-MODEL program (https://swissmodel.expasy.org/interactive; accessed on 29 April 2021). FoldX (http://foldxsuite.crg.eu/products#foldx; accessed on 29 April 2021) was used to repair the missing loops in the mutant and the wild-type structures of PBP2a and to optimize two PBP2a structures for energy minimization and the removal of amino acid side chain clashes as previously described [39]. The optimized mutant and the wild-type structures of PBP2a were then used for protein-ligand docking using AutoDock Vina [40]. The ligand cefoxitin was extracted and obtained from the structure (PBD ID, 3MZE) of the complex of cefoxitin with penicillin-binding proteins. The cefoxitin was docked with the optimized structures of wild-type and mutant PBP2a. The protein-ligand docking grid boxes in two PBP2a structures were selected based on the binding residues reported previously [24]. The binding affinity of cefoxitin with PBP2a was performed using AutoDock Vina. In AutoDock Vina, the parameter of the binding free energy has been used to determine the binding affinity of ligand (cefoxitin) with protein (PBP2a) [41]. The lesser the binding free energy, the higher the binding affinity the ligand has for the protein. AutoDock Vina is a free open-source package (http://vina.scripps.edu/; accessed on 29 April 2021) that can accurately and rapidly determine the binding affinity. AutoDock Vina is widely used with more than 6000 citations during the last ten years. AutoDock Vina has been available since 2010. AutoDock Vina is completely empirical and is comprised of hydrogen bonds, repulsion, Gaussian steric interactions, and hydrophobic and torsion terms [40]. AutoDock Vina was designed with parallel computing capabilities and is easy to use [41]. It was indicated that AutoDock Vina is accurate when determining binding affinity based on the CASF-2013 benchmark [42].

## Figures and Tables

**Figure 1 pharmaceuticals-14-00420-f001:**
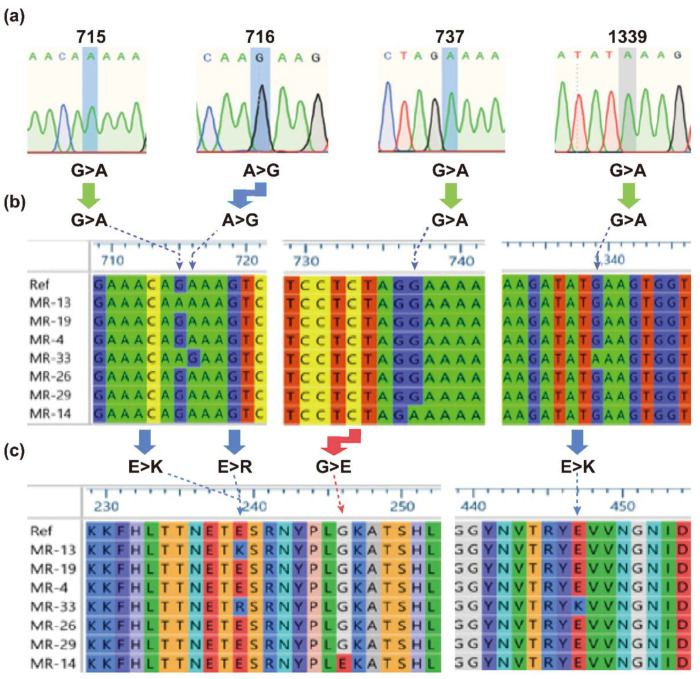
The detected variations in the *mecA* gene in the MRSA isolates (MR-13, -19, -4, -33, -26, -29, and -14). (**a**) Chromatograms showing nucleotide substitutions at different locations. (**b**) Multiple DNA sequence alignment. (**c**) Multiple protein sequence alignment. The sign “>” represents a substitution mutation. The left “E>K”, “E>R”, “G>E”, and the right “E>K” refer to E239K, E239R, G246E, and E447K, as shown by the dashed arrows. The left “G>A”, “A>G”, and “G>A”, and the right “G>A” refer to G715A, A716G, G737A, and G1339A. “Ref” represents the reference sequence (GenBank accession no. NC_002745). The MRSA isolates (MR-13, -14, and -33) show amino acid substitution(s) in PBP2a.

**Figure 2 pharmaceuticals-14-00420-f002:**
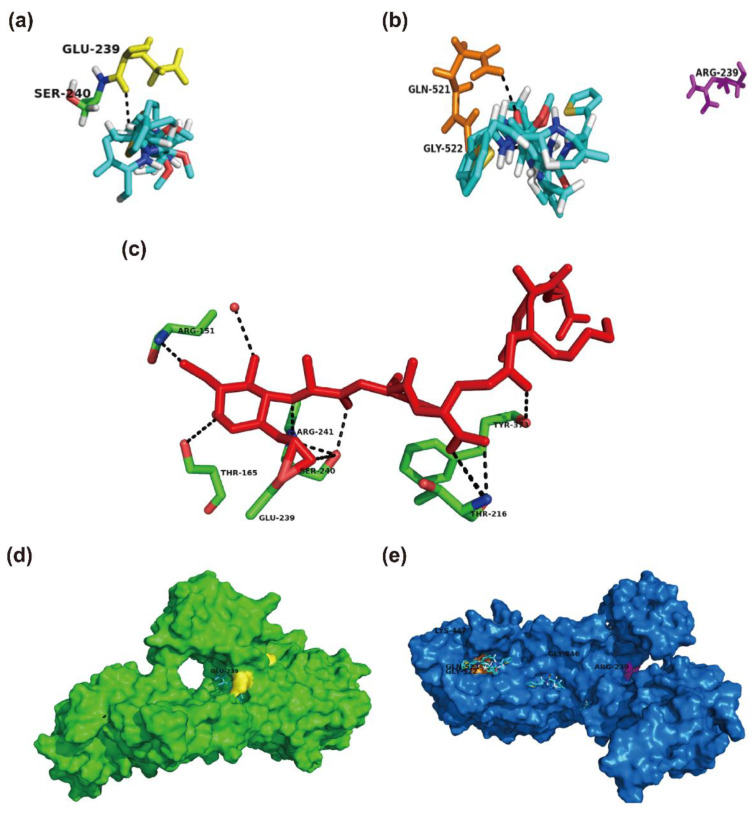
The protein-ligand docking of cefoxitin with the wild-type PBP2a and E239R-mutant PBP2a of MRSA. (**a**) Cefoxitin (shown in capped sticks with carbon in cyan, oxygen in red, nitrogen in blue, and sulfur in yellow) interacts with the Glu-239 (E239, yellow) of the wild-type PBP2a. The docked ligand trajectory shows that cefoxitin is located in close proximity to E239. (**b**) Cefoxitin interacts with Gln-521 (orange) and Gly-522 (orange) instead of Arg-239 (R239, pink) in the E239R-mutant PBP2a. (**c**) Peptidoglycan (red, GlcNAc-MurNAc pentapeptide) interacts with E239 and other residues of the wild-type PBP2a. (**d**) Surface representation of the wild-type PBP2a (green) showing that the ligand (shown in caped sticks) is positioned at the allosteric site (yellow). (**e**) The ligand (shown in capped sticks) is pointed away from the allosteric site (pink) in the E239R-mutant PBP2a. All of the interactions are shown as black dashed lines. Glu, glutamic acid; Ser, serine; Gln, glutamine; Gly, glycine; Arg, arginine; Thr, threonine; Tyr, tyrosine; Lys, lysine.

**Table 1 pharmaceuticals-14-00420-t001:** The antimicrobial disk susceptibility tests with cefoxitin 30 µg disks.

Staphylococcus aureus Isolate	Zone Diameter (mm)	Interpretive Category ^a^
MR-1	13	R
MR-2	20	R
MR-3	13	R
MR-4	13	R
MR-5	16	R
MR-6	13	R
MR-7	13	R
MR-8	17	R
MR-9	16	R
MR-10	15	R
MR-11	20	R
MR-12	19	R
MR-13	4	R
MR-14	4	R
MR-15	34	S
MR-16	13	R
MR-17	16	R
MR-18	13	R
MR-19	14	R
MR-20	23	S
MR-21	17	R
MR-22	17	R
MR-23	18	R
MR-24	16	R
MR-25	15	R
MR-26	14	R
MR-27	18	R
MR-28	29	S
MR-29	17	R
MR-30	14	R
MR-31	16	R
MR-32	17	R
MR-33	3	R

^a^ The inhibition zone diameters were interpreted using the Clinical Laboratory Standards Institute (CLSI) breakpoints [20].

**Table 2 pharmaceuticals-14-00420-t002:** The missense mutations of PBP2a associated with the resistance to cefoxitin.

Site/Source	B	N	W
MRSA isolate	MR-13	MR-14	MR-33
Amino acid change(s) of PBP2a	E239K	G246E	**E239R** E447K
Cefoxitin MIC (mg/L)	64	64	128

B, blood; N, nasal swab; W, wound sample. MRSA, methicillin-resistant *Staphylococcus aureus*; PBP2a, penicillin-binding protein 2a; MIC, minimum inhibitory concentration. **E239R** is the novel mutation of PBP2a.

**Table 3 pharmaceuticals-14-00420-t003:** The primers used for PCR amplification and DNA sequencing.

Primer Name	Primer Sequence (5′→3′)	Reference
Fnuc	GCGATTGATGGTGATACGGTT	This study
Rnuc	AGCCAAGCCTTGACGAACTAAAGC	This study
16sF	AGAGTTTGATCCTTGGCTAG	This study
16sR	GCYTACCTTGTTACGACTT	This study
mecA_DF	AAAATCGATGGTAAAGGTTGGC	This study
mecA_DR	AGTTCTGCAGTACCGGATTTGC	This study
MF	AACCGAAGAAGTCGTGTCAG	This study
MR	CATCGTTACGGATTGCTTCG	
MecAR4	GATACATTCTTTGGAACGATG	[36]
MecAF5	ACAAGATGATACCTTCGTTCCACTT	
MecAF3	GAAGATGGCTATCGTGTCAC	
MecAF4	GGTAATATCGACTTAAAACAAG	

## Data Availability

The data are contained within the article or Appendix A.

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
