# Peer review of "Mutation-Based Antibiotic Resistance Mechanism in Methicillin-Resistant Staphylococcus aureus Clinical Isolates"

_pharmaceuticals, 2021, doi:10.3390/ph14050420_

Round 1

Reviewer 1 Report

Abstract:

please pay attention and apply the changes to the abstract, which were requested in the main text!

„MRSA was first reported in the United Kingdom in 1961 and quickly disseminated in the world widely” Please rephrase

L34-L38:

Please consider including the following reference:

https://pubmed.ncbi.nlm.nih.gov/31052511/

Please consider that PBP2c and PBP2x may also lead to methicillin-resistance. Indicate this in the introduction and throughout the manuscript.

„by almost all β-lactam antibiotics” please define that 5th generation cephalosporins are still effective

„…in concert with…” Please rephrase.

2.1. Please precisely provide the origin of all isolates (i.e. isolation site)

Please provide a brief summary of the clinical problem with MRSA infections, including the association with other resistance determinants (either with chromosomal or plasmid-mediated) and the additional issue with biofilm-formation.

Please provide a brief, clinically-relevant conclusions regarding the results of the study, how these findings may impact current research/diagnostic practices.

The methdology was adequately described.

Author Response

April 15, 2021

Dear Reviewer 1, Pharmaceuticals:

I appreciate the comments that you have made regarding our manuscript (Manuscript ID: pharmaceuticals-1191809). I have carefully read your comments and the manuscript has been rewritten in response to these comments as detailed below.

The amended parts were represented and highlighted in yellow color in the revised manuscript (pharmaceuticals-1191809-for rd1.docx).

Responses to comments of Reviewer 1

  1. Comment: Please pay attention and apply the changes to the abstract, which were requested in the main text!

Response: According to your comment, I have changed the “Abstract” section, shown in the highlighted (yellow) parts of the revised manuscript (page 1).

  1. Comment: „MRSA was first reported in the United Kingdom in 1961 and quickly disseminated in the world widely” Please rephrase.

Response: The sentence was changed to the part “MRSA isolate first emerged in the United Kingdom in 1961 and quickly spread globally.” in the revised manuscript (page 2).

  1. Comment: L34-L38: Please consider including the following reference: https://pubmed.ncbi.nlm.nih.gov/31052511/

Response: The reference (Gajdács M. The continuing threat of methicillin-resistant Staphylococcus aureus. Antibiotics. 2019, 8, 52) was inserted in the revised manuscript (pages 2 and 10).

  1. Comment: Please consider that PBP2c and PBP2x may also lead to methicillin-resistance. Indicate this in the introduction and throughout the manuscript.

Response: The following sentences (or parts) were inserted in the revised manuscript:

Page 2, lines 54-57: Currently, thirteen allotypes of SCCmec (namely type I–XIII) have been defined. Among thirteen SCCmec allotypes, the only rare allotype (type XI) contains the mecC instead of mecA gene [3]. Furthermore, PBP2x shows the cefuroxime resistance in Streptococcus pneumoniae [10] (page 2).

Page 7, lines 226-227: which is consistent with the structural data of S. pneumoniae PBP2x [10] (page 7)

  1. Comment: „by almost all β-lactam antibiotics” please define that 5th generation cephalosporins are still effective

Response: According to the comment of Reviewer 4, the part “(PBP2a), which has a low affinity for β-lactam antibiotics [7]. PBP2a, a monofunctional transpeptidase, cannot be inhibited by almost all β-lactam antibiotics” was changed into “(PBP2a) [6]. PBP2a (a monofunctional transpeptidase) has a low affinity for β-lactam antibiotics and thus cannot be inhibited by these antibiotics [6]” in the revised manuscript (page 2).

  1. Comment: „…in concert with…” Please rephrase.

Response: The part “in concert with” was changed into “in cooperation with” in the revised manuscript (page 2).

  1. Comment: 2.1. Please precisely provide the origin of all isolates (i.e. isolation site)

Response: The part “(isolation sites: Main Bouleward, Zarar Shaheed Road, Lahore Road, and Link Road)” was inserted in the revised manuscript (page 3).

  1. Comment: Please provide a brief summary of the clinical problem with MRSA infections, including the association with other resistance determinants (either with chromosomal or plasmid-mediated) and the additional issue with biofilm-formation.

Response: The parts “MRSA clinical isolates became resistant to a number of antibiotic classes (e.g., fluoroquinolones, macrolides, aminoglycosides, clindamycin) as well as β-lactam antibiotics [3]. These additional resistance mechanisms are due to mutations and acquired resistance determinants, along with the formation of biofilm. Therefore, MRSA isolates are multidrug-resistant (MDR) pathogens, which is a major cause of mortality and morbidity globally [3].” were inserted in the revised manuscript (page 2).

  1. Comment: Please provide a brief, clinically-relevant conclusions regarding the results of the study, how these findings may impact current research/diagnostic practices.

Response: The parts “the CDF method along with PCR amplification of the mecA gene may be a sensitive technique for the detection of MRSA infections” and “These results can be useful both in the understanding of the development of antibiotic resistance and also design of new anti-MRSA antibiotics.” were inserted in the revised manuscript (page 8).

I appreciate you for the time in reviewing our manuscript.

With best wishes,

Prof. Sang Hee Lee

_______________________________________________

National Leading Research Laboratory

Department of Biological Sciences

Myongji University

116 Myongjiro, Yongin

Gyeonggido 17058, Republic of Korea

TEL: +82-31-330-6195

FAX: +82-31-335-8249

e-mail: sangheelee@mju.ac.kr

Reviewer 2 Report

The manuscript describes a study of 33 methicillin resistant S.aureus isolates from clinical samples from Lahore, Pakistan. The identity of the isolates was confirmed by PCR for the nuc gene and 16s ribotyping. Of these isolates 30 were resistant to cefoxitin and were mecA-positive. MRSA isolates were more likely to have come from wounds. Sequence analysis of the mecA gene from the isolates revealed variants with four amino acid substitutions (E239K, 24 E239R, G246E, and E447K). The E239R mutation has not been previously reported. In silico protein-ligand docking analysis showed that the E239R mutation in the allosteric site of PBP2a (encoded by mecA) results in conformational changes to the active site, interfering in interactions with cefoxitin. Mutation in the allosteric site of PBP2a provides allows it to function in peptidoglycan biosynthesis even in the presence of cefoxitin.

The manuscript is well written and provides novel data, which should be useful both in the understanding of the development of antibiotic resistance and also design of new antibiotics that can be used to treat recalcitrant bacterial infections.

I have the following specific comments:

Abstract

Line 20
The methicillin-resistant .. -> Methicillin-resistant

Line 21 and elsewhere
Don’t start sentence with a numeral

Introduction

Line 35
quickly disseminated in the world widely -> quickly spread world-wide.

Line 64
thus inhibits -> thus inhibiting

Discussion

Line 176
through amplifying nuc gene ->through amplifying the nuc gene

Line 184
22% clinical isolates -> 22% of clinical isolates

Line 185
affectivity -> effectiveness

Author Response

April 15, 2021

Dear Reviewer 2, Pharmaceuticals:

I appreciate the comments that you have made regarding our manuscript (Manuscript ID: pharmaceuticals-1191809). I have carefully read your comments and the manuscript has been rewritten in response to these comments as detailed below.

The amended parts were represented and highlighted in yellow color in the revised manuscript (pharmaceuticals-1191809-for rd1.docx).

Responses to comments of Reviewer 2

  1. Comment: Line 20

The methicillin-resistant .. -> Methicillin-resistant

Response: The part “The methicillin-resistant” was changed into “Methicillin-resistant” in the revised manuscript (page 1).

  1. Comment: Line 21 and elsewhere

Don’t start sentence with a numeral

Response: The part “30 MRSA isolates” was changed into “All MRSA isolates” in the revised manuscript (page 9).

The part “30 out of a total of 33 S. aureus isolates” was changed into “Of 33 S. aureus, 30 isolates” in the revised manuscript (page 3).

The part “Three out of 30 MRSA isolates” was changed into “Of 30 MRSA, three isolates” in the revised manuscript (page 3).

The part “Three MRSA isolates” was changed into “MRSA isolates” in the revised manuscript (page 4, Figure 1 legend).

The part “Three amino acid substitutions” was changed into “The amino acid substitutions” in the revised manuscript (page 5).

The part “Three mutations” was changed into “The mutations” in the revised manuscript (page 5).

  1. Comment: Line 35

quickly disseminated in the world widely -> quickly spread world-wide.

Response: The part “quickly disseminated in the world widely” was changed into “quickly spread globally” in the revised manuscript (page 2).

  1. Comment: Line 64

thus inhibits -> thus inhibiting

Response: The part “thus inhibits” was changed into “thus inhibiting” in the revised manuscript (page 2).

  1. Comment: Line 176

through amplifying nuc gene ->through amplifying the nuc gene

Response: The part “through amplifying nuc gene” was changed into “through amplifying the nuc gene” in the revised manuscript (page 6).

  1. Comment: Line 184

22% clinical isolates -> 22% of clinical isolates

Response: The part “22% clinical isolates” was changed into “22% of clinical isolates” in the revised manuscript (page 7).

  1. Comment: Line 185

affectivity -> effectiveness

Response: The part “affectivity” was changed into “effectiveness” in the revised manuscript (page 7).

I appreciate you for the time in reviewing our manuscript.

With best wishes,

Prof. Sang Hee Lee

_______________________________________________

National Leading Research Laboratory

Department of Biological Sciences

Myongji University

116 Myongjiro, Yongin

Gyeonggido 17058, Republic of Korea

TEL: +82-31-330-6195

FAX: +82-31-335-8249

e-mail: sangheelee@mju.ac.kr

Reviewer 3 Report

Sequence variation analysis of the mecA gene showed four amino acid substitutions. Novel mutation was found, which provides favourable conformation to PBP2a for its function in peptidoglycan biosynthesis even in the presence of cefoxitin. Authors investigated the genetic polymorphism of the mecA gene in clinical MRSA isolates determined the probable role of the observed novel mutation in the allosteric site of PBP2a in antibiotic resistance by protein modeling and protein-ligand docking. The Authors showed the mechanism of the high level of cefoxitin resistance.

This work is well-designed, well-performed and well-presented. My decision regarding your work is "Accept after minor revision".

Line 82-83 – S. aureus commonly …… - this sentence should be deleted.

Line 98 – Instead “The current study also showed the prevalence of MRSA from different clinical samples” should be “The prevalence of MRSA from different clinical samples was different”.

Line 100-102 – “Similar studies from Pakistan …as normal flora” this sentence should be transferred to Discussion.

Characteristics of MRSA isolates - In this section of manuscript should be included only own results, without comparison with the results of others.

Line 189 – Delete “(Table 2)”

Lines 193, 195 – Delete “(Table 2)”

Line 210 - Delete “(Table 1)”

Line 303 – “PBP2a is…..” This sentence should be deleted.

Author Response

April 15, 2021

Dear Reviewer 3, Pharmaceuticals:

I appreciate the comments that you have made regarding our manuscript (Manuscript ID: pharmaceuticals-1191809). I have carefully read your comments and the manuscript has been rewritten in response to these comments as detailed below.

The amended parts were represented and highlighted in yellow color in the revised manuscript (pharmaceuticals-1191809-for rd1.docx).

Responses to comments of Reviewer 3

  1. Comment: Line 82-83 – S. aureus commonly …… - this sentence should be deleted.

Response: The sentence “S. aureus commonly resides on the skin and in the nasal cavity of 30% of healthy people and may cause life threatening conditions” and the relevant reference were deleted in the revised manuscript (pages 3 and 11).

  1. Comment: Line 98 – Instead “The current study also showed the prevalence of MRSA from different clinical samples” should be “The prevalence of MRSA from different clinical samples was different”.

Response: The sentence “The current study also showed the prevalence of MRSA from different clinical samples” was changed into “The prevalence of MRSA from different clinical samples was different” in the revised manuscript (page 3).

  1. Comment: Line 100-102 – “Similar studies from Pakistan …as normal flora” this sentence should be transferred to Discussion.

Characteristics of MRSA isolates - In this section of manuscript should be included only own results, without comparison with the results of others.

Response: The sentences “Similar studies from Pakistan and India have reported the highest prevalence of MRSA from wound specimens  [26, 27]. The highest frequency of MRSA from wounds could be due to its presence on the skin as normal flora. The frequency of MRSA from blood specimens was 23% (n=7), which is greater than the previously reported study showing 12% MRSA in blood specimens, following wound source (26%) [28]. The frequency of MRSA from nasal swabs was 10% (n=3), which is in good agreement with the previously reported data from Pakistan [29]. The frequency of MRSA from ear swabs was 7% (n=2), which is lower than the frequencies (27-41%) reported previously [29, 30]. However, the sample size was so small that we were not able to draw a clear epidemiological aspect of MRSA showing the resistance to cefoxitin in Pakistan.” were transferred to the “Discussion” section in the revised manuscript (page 7).

And the sentence “Furthermore, the frequency of MRSA from blood specimens, nasal swabs, and ear swabs was 23% (n=7), 10% (n=3), and 7% (n=2)” was inserted in the revised manuscript (page 3).

  1. Comment: Line 189 – Delete “(Table 2)”

Response: The part “(Table 2)” was deleted in the revised manuscript (page 7).

  1. Comment: Lines 193, 195 – Delete “(Table 2)”

Response: The parts “(Table 2)” and “, Table 2” was deleted in the revised manuscript (page 7).

  1. Comment: Line 210 - Delete “(Table 1)”

Response: The part “(Table 1)” was deleted in the revised manuscript (page 7).

  1. Comment: Line 303 – “PBP2a is…..” This sentence should be deleted.

Response: The sentence “PBP2a is produced by MRSA” and the relevant reference were deleted in the revised manuscript (pages 9 and 12).

I appreciate you for the time in reviewing our manuscript.

With best wishes,

Prof. Sang Hee Lee

_______________________________________________

National Leading Research Laboratory

Department of Biological Sciences

Myongji University

116 Myongjiro, Yongin

Gyeonggido 17058, Republic of Korea

TEL: +82-31-330-6195

FAX: +82-31-335-8249

e-mail: sangheelee@mju.ac.kr

Reviewer 4 Report

The manuscript entitled "Novel mecA variant conferring antibiotic resistance in clinical methicillin-resistant Staphylococcus aureus isolates" aimed to elucidate the mutation-based antimicrobial resistance mechanisms in methicillin-resistant Staphylococcus aureus. I found that this paper is interesting but still suffers from poor scientific writing. The title is unclear and incorrect. Both abstract and text sections have a critical problem with sense and flow. Many typo errors can be found throughout. The authors separate the Results and Discussion, but the Results section still contains several discussions and literature cited. Tables and Figures should not contain undefined abbreviations. The paper needs significant revision with support from an English scientific proofread service. Some examples of my comments can be found below.

Line 17: What is PBP2a?

Lines 17-18: What is the natural function (peptidoglycan biosynthesis of binding to b-lactam antibiotics) of PBR2a? Please revise with the proper order.

Lines 19-20: Resistance mechanism of what and to what?

Lines 20-21: Is this sentence giving any sense?

Line 21: Please revise, do not start a sentence with a number.

Line 22: Why cefoxitin, and how this antibiotic links to the mecA gene? Please specify why this b-lactam antibiotic was used? Also, confirmed what?

Line 23: Did you mean MRSA in general or your MRSA isolates?

Line 28: What do you mean by favorable? How the mutated PBP2a works in comparison to the wild-type one? 

Line 35: It seems that "in the world" is sufficient. Then, please remove "widely"?

Lines 38-39: Redundancy. Please revise or remove.

Line 40: ... acquires .... by acquiring ... ? Please revise. Also, "additional" of what?

Lines 40-42: Revise its flow and sense.

Line 58: mecA in italic.

Line 92: considered no "as."

Line 93: -33?

Line 95: Full-term of MICs.

Lines 100-102:  Go to the discussion.

Line 114: mecA in no italic.

Author Response

April 15, 2021

Dear Reviewer 4, Pharmaceuticals:

I appreciate the comments that you have made regarding our manuscript (Manuscript ID: pharmaceuticals-1191809). I have carefully read your comments and the manuscript has been rewritten in response to these comments as detailed below.

The amended parts were represented and highlighted in yellow color in the revised manuscript (pharmaceuticals-1191809-for rd1.docx).

Responses to comments of Reviewer 4

  1. Comment: The title is unclear and incorrect.

Response: The title was changed into “Mutation-Based Antibiotic Resistance Mechanism in Methicillin-Resistant Staphylococcus aureus Clinical Isolates” in the revised manuscript (page 1).

  1. Comment: Both abstract and text sections have a critical problem with sense and flow. Many typo errors can be found throughout.

Response: Many parts were changed as follows:

(1) Line 17: What is PBP2a?

Response: The part “PBP2a” was changed into “penicillin-binding protein 2a (PBP2a)” in the revised manuscript (page 1).

(2) Lines 17-18: What is the natural function (peptidoglycan biosynthesis or binding to b-lactam antibiotics) of PBR2a? Please revise with the proper order.

Response: The part “PBP2a exhibits poor affinity towards β-lactam antibiotics and participates in peptidoglycan bio-synthesis” was changed into “PBP2a participates in peptidoglycan biosynthesis and exhibits poor affinity towards β-lactam antibiotics” in the revised manuscript (page 1).

(3) Lines 19-20: Resistance mechanism of what and to what?

Response: The part “resistance mechanism” was changed into “antibiotic resistance mechanism” in the revised manuscript (page 1).

(4) Lines 20-21: Is this sentence giving any sense?

Response: The sentence “The methicillin-resistant Staphylococcus aureus (MRSA) isolates from clinical samples were identified through both phenotypic and genotypic characteristics” was changed into “Methicillin-resistant Staphylococcus aureus (MRSA) isolates from clinical samples were identified using phenotypic and genotypic techniques” in the revised manuscript (page 1).

(5) Line 21: Please revise, do not start a sentence with a number.

Response: The part “30 MRSA isolates” was changed into “All MRSA isolates” in the revised manuscript (page 9).

The part “30 out of a total of 33 S. aureus isolates” was changed into “Of 33 S. aureus, 30 isolates” in the revised manuscript (page 3).

The part “Three out of 30 MRSA isolates” was changed into “Of 30 MRSA, three isolates” in the revised manuscript (page 3).

The part “Three MRSA isolates” was changed into “MRSA isolates” in the revised manuscript (page 4, Figure 1 legend).

The part “Three amino acid substitutions” was changed into “The amino acid substitutions” in the revised manuscript (page 5).

The part “Three mutations” was changed into “The mutations” in the revised manuscript (page 5).

(6) Line 22: Why cefoxitin, and how this antibiotic links to the mecA gene? Please specify why this b-lactam antibiotic was used? Also, confirmed what?

Response: The part “30 MRSA isolates showed resistant against cefoxitin, which were further confirmed through molecular diagnostics targeting the mecA gene” was deleted in the revised manuscript (page 1). This part shows redundancy due to the previous sentence “Methicillin-resistant Staphylococcus aureus (MRSA) isolates from clinical samples were identified using phenotypic and genotypic techniques.”

(7) Line 23: Did you mean MRSA in general or your MRSA isolates?

Response: The part “(60%)” was changed into “(60%, 18 out of 30)” in the revised manuscript (page 1).

(8) Line 28: What do you mean by favorable? How the mutated PBP2a works in comparison to the wild-type one?

Response: The part “Thus, the present report indicated that mutation in the allosteric site of PBP2a provides favorable conformation to PBP2a for its function in peptidoglycan biosynthesis even in the presence of cefoxitin” was changed into “Therefore, the present report indicated that mutation in the allosteric site of PBP2a provides a more closed active site conformation than wild-type PBP2a and then causes the high-level resistance to cefoxitin” in the revised manuscript (page 1).

(9) Line 35: It seems that "in the world" is sufficient. Then, please remove "widely"?

Response: According to the comment of Reviewer 1, the part “quickly disseminated in the world widely” was changed into “quickly spread globally” in the revised manuscript (page 1).

(10) Lines 38-39: Redundancy. Please revise or remove.

Response: The part “The prevalence of antibiotic-resistant S. aureus has been increasing all over the world due to its potential of gaining resistance against a broad range of antibiotics” and relevant references were deleted in the revised manuscript (pages 2 and 10).

(11) Line 40: ... acquires .... by acquiring ... ? Please revise. Also, "additional" of what?

Response: The part “by acquiring an additional penicillin-binding protein 2a (PBP2a)” was changed into “by a penicillin-binding protein 2a (PBP2a)” in the revised manuscript (page 2).

(12) Lines 40-42: Revise its flow and sense.

Response: The part “(PBP2a), which has a low affinity for β-lactam antibiotics [7]. PBP2a, a monofunctional transpeptidase, cannot be inhibited by almost all β-lactam antibiotics” was changed into “(PBP2a) [6]. PBP2a (a monofunctional transpeptidase) has a low affinity for β-lactam antibiotics and thus cannot be inhibited by these antibiotics [6]” in the revised manuscript (page 2).

(13) Line 58: mecA in italic.

Response: The part “mecA” was changed into “mecA” in the revised manuscript (page 2).

(14) Line 92: considered no "as."

Response: The part “considered as” was changed into “considered” in the revised manuscript (page 3).

(15) Line 93: -33?

Response: The part “33” was changed into “-33” in the revised manuscript (page 3).

(16) Line 95: Full-term of MICs.

Response: The part “MICs” was changed into “minimum inhibitory concentrations (MICs)” in the revised manuscript (page 3).

The part “minimum inhibitory concentration (MIC)” was changed into “MIC” in the revised manuscript (page 9).

(17) Lines 100-102:  Go to the discussion.

Response: According to the comment of Reviewer 3, the sentences “Similar studies from Pakistan and India have reported the highest prevalence of MRSA from wound specimens  [26, 27]. The highest frequency of MRSA from wounds could be due to its presence on the skin as normal flora. The frequency of MRSA from blood specimens was 23% (n=7), which is greater than the previously reported study showing 12% MRSA in blood specimens, following wound source (26%) [28]. The frequency of MRSA from nasal swabs was 10% (n=3), which is in good agreement with the previously reported data from Pakistan [29]. The frequency of MRSA from ear swabs was 7% (n=2), which is lower than the frequencies (27-41%) reported previously [29, 30]. However, the sample size was so small that we were not able to draw a clear epidemiological aspect of MRSA showing the resistance to cefoxitin in Pakistan.” were transferred to the “Discussion” section in the revised manuscript (page 7).

And the sentence “Furthermore, the frequency of MRSA from blood specimens, nasal swabs, and ear swabs was 23% (n=7), 10% (n=3), and 7% (n=2)” was inserted in the revised manuscript (page 3).

(18) Line 114: mecA in no italic.

Response: The part “mecA” was changed into “mecA” in the revised manuscript (page 4).

  1. Comment: The authors separate the Results and Discussion, but the Results section still contains several discussions and literature cited.

Response: According to the comments of you and Reviewer 3, we separated the Results and Discussion as fellows.

The sentences “Similar studies from Pakistan and India have reported the highest prevalence of MRSA from wound specimens  [26, 27]. The highest frequency of MRSA from wounds could be due to its presence on the skin as normal flora. The frequency of MRSA from blood specimens was 23% (n=7), which is greater than the previously reported study showing 12% MRSA in blood specimens, following wound source (26%) [28]. The frequency of MRSA from nasal swabs was 10% (n=3), which is in good agreement with the previously reported data from Pakistan [29]. The frequency of MRSA from ear swabs was 7% (n=2), which is lower than the frequencies (27-41%) reported previously [29, 30]. However, the sample size was so small that we were not able to draw a clear epidemiological aspect of MRSA showing the resistance to cefoxitin in Pakistan.” were transferred to the “Discussion” section in the revised manuscript (page 7).

And the sentence “Furthermore, the frequency of MRSA from blood specimens, nasal swabs, and ear swabs was 23% (n=7), 10% (n=3), and 7% (n=2)” was inserted in the revised manuscript (page 3).

  1. Comment: Tables and Figures should not contain undefined abbreviations.

Response: Abbreviations were defined in Tables and Figures as fellows.

The part “CLSI” was changed into “Clinical Laboratory Standards Institute (CLSI)” in the revised manuscript (page 4, Table 1).

The part “The left “G>A”, “A>G”, “G>A”, and the right “G>A” refer to G715A, A716G, G737A, and G1339A” was inserted in the revised manuscript (page 4, Figure 1).

The part “MRSA, methicillin-resistant Staphylococcus aureus; PBP2a, penicillin-binding protein 2a; MIC, minimum inhibitory concentration” was inserted in the revised manuscript (page 5, Table 2).

The part “Glu, glutamic acid; Ser, serine; Gln, glutamine; Gly, glycine; Arg, arginine; Thr, threonine; Tyr, tyrosine; Lys, lysine” was inserted in the revised manuscript (page 6, Figure 2).

I appreciate you for the time in reviewing our manuscript.

With best wishes,

Prof. Sang Hee Lee

_______________________________________________

National Leading Research Laboratory

Department of Biological Sciences

Myongji University

116 Myongjiro, Yongin

Gyeonggido 17058, Republic of Korea

TEL: +82-31-330-6195

FAX: +82-31-335-8249

e-mail: sangheelee@mju.ac.kr

This manuscript is a resubmission of an earlier submission. The following is a list of the peer review reports and author responses from that submission.